# Design and Experimental Study of a Hybrid Micro-Vibration Isolation System Based on a Strain Sensor for High-Precision Space Payloads

**DOI:** 10.3390/s24051649

**Published:** 2024-03-03

**Authors:** Qiwei Guo, Jian Zhou, Liang Li, Minglong Xu, Guoan Tang

**Affiliations:** 1Department of Aeronautics and Astronautics, Fudan University, Shanghai 200433, China; 21110290021@m.fudan.edu.cn (Q.G.); tangguoan@fudan.edu.cn (G.T.); 2Shanghai Aerospace System Engineering Institute, Shanghai 201108, China; 3State Key Laboratory for Strength and Vibration of Mechanical Structures, School of Aerospace Engineering, Xi’an Jiaotong University, Xi’an 710049, China; liliang870130@stu.xjtu.edu.cn (L.L.); mlxu@xjtu.edu.cn (M.X.)

**Keywords:** micro-vibration, active–passive isolator, vibration control, space payload, strain sensor

## Abstract

Micro-vibrations significantly influence the imaging quality and pointing accuracy of high-precision space-borne payloads. To mitigate this issue, vibration isolation technology must be employed to reduce the transmission of micro-vibrations to payloads. In this paper, a novel active–passive hybrid isolation (APHI) system based on a strain sensor is proposed for high-precision space payloads, and corresponding theoretical and experimental studies are implemented. First, a theoretical analysis model of the APHI system is established using a two-degrees-of-freedom system, and an integral control method based on strain sensing is presented. Then, an electromagnetic damper, active piezoelectric actuator, and strain sensor are designed and manufactured. Finally, an APHI experimental system is implemented to validate the effectiveness of electromagnetic damping and strain-sensing active control. Additionally, the control effects of acceleration, displacement, and strain sensors are compared. The results demonstrate that strain sensors can achieve effective active damping control, and the control method based on strain sensors can effectively suppress the payload response while maintaining stability. Both displacement and strain sensors exhibit superior suppression effects compared with the acceleration sensor, with the strain sensor showing greater potential for practical engineering applications than the displacement sensor.

## 1. Introduction

With the rapid development of space technology, future space payloads are expected to have higher resolution and higher pointing accuracy. For instance, the James Webb Space Telescope requires milliarcsecond-level pointing accuracy [1], while laser communication satellites also require extremely high pointing accuracy and attitude stability for precise signal transmission [2]. However, the micro-vibration generated by the moving components on the satellite, such as the rotating momentum wheel and flywheel, can significantly impact the imaging quality and pointing accuracy of the payload when it is transmitted to the payload [3]. Therefore, micro-vibration isolation technology is needed to reduce the transmission of micro-vibration to payloads.

Vibration isolation technology aims to block the transmission of micro-vibration to space payloads, primarily to minimize its adverse effects on sensitive optical equipment. It encompasses various approaches, including passive isolation technology, semi-active isolation technology, active isolation technology, and active–passive hybrid isolation technology. Passive vibration isolation technology [4,5,6] is currently the most mature and widely used method, offering advantages such as a simple structure, high stability and reliability, easy maintenance, and the nonrequirement of an external energy supply. However, passive isolation largely relies on its stiffness and damping characteristics, and it cannot suppress low-frequency vibrations; it may even amplify the vibration response within the frequency band range $\left[\sqrt{2}\omega_{n}/2,\sqrt{2}\omega_{n}\right]$ [7]. Semi-active isolation technology [8,9,10] can be considered an adaptive form of passive isolation that requires external energy to modify stiffness and damping. However, it still has limitations similar to those of passive isolation. Active vibration isolation technology [11,12,13,14] can overcome the shortcomings of passive and semi-active isolation by effectively suppressing low-frequency vibrations. However, active isolation primarily focuses on low-frequency suppression. In contrast, active–passive hybrid isolation technology [15,16,17] combines the strengths of both active and passive isolation methods, addressing the drawbacks of each and achieving optimal isolation performance. In an active–passive hybrid isolation system, passive isolation primarily handles high-frequency isolation, while active isolation is primarily responsible for low-frequency isolation. Additionally, in the event of active isolation system failure, passive isolation continues to provide isolation, thereby enhancing the stability of the overall isolation system. For space payloads that require high resolution and pointing accuracy, active–passive hybrid isolation offers a wider frequency band of isolation, effectively reducing the transmission of micro-vibrations to space-borne optical sensitive payloads.

The US Air Force Research Laboratory has developed an active–passive hybrid vibration isolation system called “SUITE”, which utilizes piezoelectric actuators [18,19]. Each strut of the system consists of three components: a piezoelectric actuator, a vibration measurement sensor, and a passive isolator. The piezoelectric actuator is responsible for active vibration suppression within a designed frequency band of 5–20 Hz. A velocity detector serves as the vibration measurement sensor, while elastic elements are employed for the passive part. SUITE, functioning as an in-orbit verification platform, incorporates DSP technology and enables the verification of various control algorithms supported by DSP, including multivariable control algorithms and adaptive control algorithms. Moreover, the US Air Force Research Laboratory has developed the VISS vibration isolation system [20], which is utilized for isolating infrared telescopes and reducing vibrations from refrigerators and satellites. VISS consists of six struts that integrate acceleration sensors and active actuators. The active actuator is implemented using a voice coil motor, and the passive part, consisting of springs and viscous fluid, is connected in parallel to the voice coil motor. The system employs a classic acceleration feedback control algorithm, and the design frequency band for isolation is 5–200 Hz. Jang et al. [21] developed an APHI system based on air springs and piezoelectric actuators. The active control method employed the LMS control algorithm based on acceleration feedback, resulting in a vibration attenuation of 70%. Nakamura et al. [22] designed an active–passive vibration isolation system that incorporated an air spring as the passive component, a giant magnetostrictive actuator as the active component, and acceleration as the feedback signal. The vibration isolation system achieved a maximum attenuation of 30 dB within the frequency range of <100 Hz. Zhang et al. [23] developed a compact active–passive integrated actuator using piezoelectric ceramics and rubber dampers. They constructed a six-dimensional orthogonal vibration isolation platform combined with a force sensor as the sensing unit. However, the initial frequency of this isolator was higher than 18 Hz, and it could attenuate the vibration response by 18 dB. According to the abovementioned literature, the passive components of active–passive hybrid isolation systems mainly consist of springs, air springs, and rubber dampers, while the active components predominantly employ voice coil motors, piezoelectric actuators, and magnetostrictive actuators. The sensing units commonly include force sensors, acceleration sensors, and velocity detectors. The Stewart parallel platform, as depicted in Figure 1, is supported by six struts and can achieve six degrees of freedom of motion. So, The Stewart structure is commonly utilized as a micro-vibration isolation platform for space payloads. When the Stewart parallel platform is decoupled, the controller becomes simpler, allowing individual control for each strut. Thus, the performance of the isolation platform primarily relies on the design of a single strut, which allows for the verification of the control strategy.

In order to achieve a smaller initial isolation frequency and a more high-quality and practical sensor, this paper presents a novel APHI single-strut structure with a strain sensor and a corresponding control method for the micro-vibration isolation platform of space payloads. The APHI unit consists of a serially connected electromagnetic damper and an active piezoelectric actuator with a strain sensor applied to the membrane as the sensing element.

## 2. Theoretical Study

### 2.1. Theoretical Model

Figure 2 illustrates the structural model of the APHI strut. The strut primarily comprises an electromagnetic damper, two membranes, an active piezoelectric actuator, and connecting rods. The electromagnetic damper and membranes form the passive part of the strut, with the fundamental frequency of the isolation system determined by the elastic stiffness of the membranes. The active part consists of a piezoelectric actuator with high axial stiffness. A strain sensor is attached to one of the membranes, enabling the measurement of membrane strain resulting from the relative motion between the base and the piezoelectric actuator.

The APHI system of the single strut is depicted in Figure 2. The base and active piezoelectric actuator are connected by an electromagnetic damper with two membranes. The membranes provide connection stiffness, and the coil of the electromagnetic damper cuts the magnetic induction line to generate electromagnetic damping. Therefore, the connection between the base and the active piezoelectric actuator can be simplified as a spring and damping connection. The active piezoelectric actuator itself can be simplified into a single-degree-of-freedom mass–spring–damping system, and it is also an actuating element that outputs the driving force. So, the active–passive hybrid isolation system can be simplified as a two-degrees-of-freedom mathematical model with a controller, as shown in Figure 3. In the figure, $m_{1}$ denotes the sum of the shell mass of the active piezoelectric actuator and the mass of the connecting components between the active piezoelectric actuator and the electromagnetic damper; $m_{2}$ denotes the mass of the payload; $c_{1}$ denotes the damping of the electromagnetic damper, including structural damping and electromagnetic damping; $c_{2}$ denotes the structural damping of the active piezoelectric actuator; $k_{1}$ denotes the total axial stiffness of the two membranes; $k_{2}$ denotes the equivalent dynamic stiffness of the active piezoelectric actuator; $x_{0}$ is the basic displacement excitation; $x_{1}$ is the displacement of $m_{1}$; $x_{2}$ is the displacement of $m_{2}$; $f_{a}$ is the dynamic output force of the active piezoelectric actuator obtained by the control algorithm. The strain sensor mainly measures the relative motion between the active piezoelectric actuator and the base; therefore, the strain can be expressed as $\beta \left(x_{1}-x_{0}\right)$, where β is the proportional coefficient between strain and relative displacement $\left(x_{1}-x_{0}\right)$.

According to Newton’s second law, the dynamic equation of the isolation system can be obtained as follows:(1)$$m_{2}{\ddot{x}}_{2}+c_{2}({\dot{x}}_{2}-{\dot{x}}_{1})+k_{2}(x_{2}-x_{1})=f_{a},$$
(2)$$m_{1}{\ddot{x}}_{1}+c_{1}({\dot{x}}_{1}-{\dot{x}}_{0})+c_{2}({\dot{x}}_{1}-{\dot{x}}_{2})+k_{1}(x_{1}-x_{0})+k_{2}(x_{1}-x_{2})={-f}_{a},$$

### 2.2. Controller Design

When $f_{a}=0$, the isolation strut is a passive isolation system. The transfer function from the base excitation to the payload can be obtained through the Laplace transform:(3)$$\frac{x_{2}(s)}{x_{0}(s)}=\frac{\left(k_{1}+c_{1}s)(k_{2}+c_{2}s\right)}{\left(m_{1}s^{2}+c_{1}s+k_{1})(m_{2}s^{2}+c_{2}s+k_{2})+m_{2}s^{2}(c_{2}s+k_{2}\right)},$$

Adopting the PID control strategy, the control force can be expressed as
(4)$$f_{a}=\left(g_{P}+g_{I}\frac{1}{s}+g_{D}s\right)\beta \left(x_{1}(s)-x_{0}(s)\right),$$
where $g_{P}$, $g_{I}$, and $g_{D}$ are the proportional coefficient, integral coefficient, and differential coefficient in the PID control strategy, respectively. In the case of APHI, the transfer function from base excitation to the payload can be expressed as follows:(5)$$\frac{x_{2}(s)}{x_{0}(s)}=\frac{-(g_{P}\beta +g_{I}\beta /s+g_{D}\beta s)m_{1}s^{2}+\left(k_{1}+c_{1}s\right)(k_{2}+c_{2}s)}{\left(m_{1}s^{2}+c_{1}s+k_{1}\right)\left(m_{2}s^{2}+c_{2}s+k_{2}\right)+m_{2}s^{2}(c_{2}s+k_{2}+g_{P}\beta +g_{I}\beta /s+g_{D}\beta s)},$$

The parameters are assumed as follows: $m_{1}=0.25$ kg, $m_{2}=22$ kg, $k_{1}=10,000$ N/m, $k_{2}=4\times 10^{8}$ N/m, $c_{2}=0$, $c_{1}=5$ N·s/m (open-loop), $c_{1}=70$ N·s/m (electromagnetic damping), and $\beta =1.6\times 10^{8}$. Figure 4 illustrates the transmission characteristics from base excitation to the payload under different conditions: open loop, electromagnetic damping, active control (P), active control (I), and active control (D). The open-loop condition represents passive isolation without electromagnetic damping. The electromagnetic damping condition represents passive isolation with electromagnetic damping. The active control (P) condition represents active–passive control with electromagnetic damping, and $g_{I}=0$, $g_{D}=0$. The active control (I) condition represents active–passive control with electromagnetic damping, and $g_{P}=0$, $g_{D}=0$. The active control (D) condition represents active–passive control with electromagnetic damping, and $g_{P}=0$, $g_{I}=0$. From Figure 4, under the open-loop condition, the system exhibits two dominant frequencies. The fundamental frequency is 3.36 Hz, with a transmission amplitude of 37.4 dB, primarily attributed to the behavior of the membranes. Owing to the high stiffness of the piezoelectric actuator, the second-order frequency is significantly higher, reaching 6400 Hz, with a transmission amplitude of only −39.8 dB. Introducing electromagnetic damping does not change the frequency values, but it reduces the transmission amplitude of the fundamental frequency to 16.7 dB. The high-frequency transmission amplitude of the system with electromagnetic damping is slightly higher than that of the open-loop condition, mainly owing to the phase change in vibration transmission characteristics caused by damping. When using active control (P), active control (I) and active control (D), the transmission amplitude at the fundamental frequency significantly decreases to a very small value. However, in active control (P) and active control (D), the transmission amplitude in the high-frequency range notably increases. Additionally, owing to the presence of system poles distributed on the right side of the zero-pole map, the control system becomes unstable, as shown in Figure 5a,c. By utilizing active control (I), the transmission amplitude at the fundamental frequency is reduced, and the high-frequency part also decreases compared with the case of electromagnetic damping alone. Furthermore, the control system remains stable, as illustrated in Figure 5b. Therefore, in this APHI system, when the strain sensor is used, the active control (I) strategy is reasonable and provides stability.

## 3. Experimental Study

### 3.1. Electromagnetic Damper Design

Figure 6 shows the structural schematic and physical diagram of the electromagnetic damper. The electromagnetic damper includes a top cap, upper cross-shaped membrane, connecting shaft, coil framework, coil, magnet steel, outer covering, lower cross-shaped membrane, and bottom cap. The coil is wound around the coil frame and then located inside the magnet steel. The magnet steel is fixed inside the outer covering. The upper and lower cross-shaped membranes fit tightly through the connecting shaft. The coil framework is fixedly connected to the connecting shaft. The top cap, upper and lower cross-shaped membranes, and bottom cap are connected to the outer covering. 

The electromagnetic damper adopts a design with the upper and lower double cross-shaped membranes, primarily serving as passive isolation and guidance components. The combination design method of a single membrane and guide bearing is not used mainly to ensure the damping effect of the electromagnetic damper. The distance between the magnetic steel and the magnetic coil is very small, ~0.5 mm. During the installation and operation of a single-membrane electromagnetic damper, it is challenging to ensure that no compression force exists between the shaft and the guide bearing. The presence of compression force would lead to mutual friction. Although this friction force is small, it cannot be overcome under micro-vibration excitation and would result in the failure of passive isolation.

The cross-shaped membrane is fabricated using stainless-steel material, which has a relatively high yield strength, and then is subjected to static analysis using finite element software, as depicted in Figure 7. Under a load of 1 N, the maximum displacement of the membrane is 0.0843 mm, so the stiffness of the single membrane can be determined as 11,862.4 N/m.

### 3.2. Active Piezoelectric Actuator Design

An active piezoelectric actuator can be designed using piezoelectric stacks. According to the inverse piezoelectric effect of piezoelectric stacks, a piezoelectric actuator can generate corresponding driving forces based on active control signals and then apply them to the controlled structure to control the movement, achieving suppression of the controlled object. Figure 8 illustrates the structural schematic and physical diagram of the active piezoelectric actuator. The main components of an active piezoelectric actuator include piezoelectric stacks, a diamond ring, a ball bearing, and an output shaft. To increase the displacement of the actuator, it is designed by splicing three piezoelectric stacks. The piezoelectric stack can only withstand axial pressure. If the local stress or bending moment on the piezoelectric ceramic becomes too high, the piezoelectric stack may be damaged. To achieve torque unloading of the actuator axis, thrust ball bearings are used. Additionally, a ball contact bending moment elimination mechanism is integrated at the upper end of the piezoelectric ceramic to prevent the transmission of bending moments to the ceramic and increase the actuator’s resistance to bending. The piezoelectric stack is nested within the diamond ring, with three piezoelectric stacks connected in series. Flexible hinges are strategically positioned on the diamond ring to improve the lateral impact resistance of the piezoelectric ceramics. The parameters of the designed piezoelectric actuator are presented in Table 1. Figure 9 shows the relationship between actuating force and voltage in the cases of 1 Hz, 5 Hz, 10 Hz, and 20 Hz. It can be seen that under an actuating frequency of less than 20 Hz, there is a slight decrease in the actuating force, which is much greater than the actuating force (2 N) required for vibration suppression in this study. In the cases of 1 Hz, 5 Hz, 10 Hz, and 20 Hz, the required powers at 100 V are 0.2 W, 1 W, 2 W, and 4 W, respectively.

### 3.3. Strain Sensor Design

To achieve high-precision measurement, sensitivity, and minimal temperature change error, the strain gauge is implemented using a Wheatstone full bridge configuration. It is bonded to the root of the cross-shaped membrane, in which the strain is most prominent during membrane deformation. The schematic depicting the strain gauge’s placement is illustrated in Figure 10a. Eight strain gauges are attached on the front and back sides of the cross-shaped membrane. The construction of the full bridge circuit for the strain gauge is shown in Figure 10b. The physical diagram of the strain sensor design is presented in Figure 11, with the strain sensor only being affixed to the upper cross-shaped membrane.

### 3.4. Experimental System Design

Figure 12 illustrates a hybrid vibration isolation experimental system consisting of a single strut. The membrane cannot bear the weight of the payload, and in order to simulate the free state of the membrane in space without gravity conditions, a horizontal suspension method is adopted to mitigate the influence of gravity. The experimental system comprises three main components: the structure, the sensor and measurement system, and control. The structural elements encompass an exciter, an electromagnetic damper, an active piezoelectric actuator, a payload, and connectors, as shown in Figure 13. To accurately simulate the micro-vibration excitation source and ensure that the actuator generating the excitation remains undisturbed by active control forces, a self-developed piezoelectric actuator with an output displacement of ±40 μm is used as the exciter. The actuator is horizontally installed on a fixed base. The electromagnetic damper is connected to the exciter to achieve passive isolation of the vibration source. The active piezoelectric actuator is connected to the electromagnetic damper via connectors, enabling active control of the system; a 22 kg mass block is used to simulate the spatial payload, and it is connected to the active piezoelectric actuator through connectors. The sensor and measurement part primarily consists of a strain gauge, accelerometer, and laser displacement meter. The control results obtained from the strain sensor are compared with those from the accelerometer and laser displacement meter for analysis and evaluation in time and frequency domain. Moreover, a laser displacement meter is utilized as a measurement tool to validate the control effectiveness. The strain sensor is affixed to the upper cross-shaped membrane, and the real-time strain-generated voltage is transmitted to the dSPACE acquisition system as an analog signal through the strain acquisition box. The acceleration is measured using a microelectromechanical system (MEMS) accelerometer, which requires a ±12 V power supply. We designed an acceleration sensor acquisition board; the collected real-time acceleration voltage signal enters the dSPACE in the form of an analog signal. The acceleration sensor is not directly installed on the payload but rather on the connection fixture between the payload and the piezoelectric actuator. This arrangement is necessitated by the fact that the actual payload is mounted on a Stewart platform with six struts. To enable individual control of each strut, the accelerometer must be placed between each strut and the payload. The real-time displacement signal from the laser displacement meter is also connected to the dSPACE via an analog signal. For the experiment, the dSPACE RTI-1102 serves as the real-time control system. The control algorithm is developed using Matlab/Simulink and then compiled into the dSPACE to facilitate the control design of the entire system. The control voltage generated by the control algorithm drives the active piezoelectric actuator through a PI power amplifier to achieve active control.

### 3.5. Experimental Results

To determine the fundamental frequency of the system, the transient response method is used. A transient impact is applied to the payload, and the displacement response of the mass block is measured using the laser displacement meter, as shown in Figure 14a. The amplitude–frequency characteristic of the system response is obtained through fast Fourier transform (FFT), as shown in Figure 14b. The fundamental frequency of the system is 5.296 Hz, which aligns with the designed frequency.

Through the analysis in Section 2.2, it can be seen that electromagnetic damping primarily plays a significant role in the resonance peak, and active control also achieves active damping using active control (I), which effectively suppress the resonance peak. Therefore, a sinusoidal signal with a fundamental frequency of 5.296 Hz is used as the vibration source for excitation, and the suppression effect of hybrid control based on the strain sensor is compared, as shown in Figure 15a. From the figure, in the open-loop state without electromagnetic damping, the peak-to-peak amplitude of the load response is 104.4 μm. Under the influence of electromagnetic damping, the peak-to-peak amplitude of the load response is 69 μm, representing a decrease of 3.5 dB. Through active control using strain sensing, the load response is further reduced to 2.4 μm, resulting in a decrease of 32.7 dB. The results show that the strain sensor control method effectively suppresses the response of the payload, and the system remains stable within a wide range of integral coefficient values during the experiment. In actual spatial payloads, the excitation received by the payload is random. Therefore, a narrowband random spectrum of 0.2–20 Hz is applied for excitation in this study. The comparison results are shown in Figure 15b. As shown in the figure, in the open-loop state without electromagnetic damping, the root mean square value of the payload displacement response is 29.9 μm. In the case of electromagnetic damping and active control, the root mean square values are 16.6 and 3.2 μm, respectively, representing reductions of 5.1 and 19.4 dB compared with the open-loop state. Based on the above experimental research, the electromagnetic damping and active control effects fully comply with the results of theoretical model analysis. This indicates that the APHI system based on the proposed strain sensor is effective theoretically and experimentally. 

Figure 16 compares the active control effects under acceleration sensor, displacement sensor, and strain sensor conditions. The control effects based on strain and displacement sensors are comparable, while the effect of the acceleration sensor is slightly inferior. As shown in Figure 16a, under sinusoidal excitation at the fundamental frequency, compared with the response amplitude controlled by strain and displacement sensors, the response amplitude controlled by the acceleration sensor is larger and more unstable, exhibiting random response behavior. A narrowband random spectrum of 0.2–20 Hz is applied for excitation. The comparison results under different sensors are shown in the time domain and frequency spectrum in Figure 16b. The root mean square values are 4.47, 3.15, and 3.18 μm in the case of acceleration, displacement, and strain sensors, respectively. The response amplitude controlled by the acceleration sensor is also larger. Additionally, when the acceleration sensor is used, the low-frequency part is prone to an increase in amplitude, whereas strain and displacement sensing do not exhibit this phenomenon. In order to remove the direct-current (DC) signal, a high-pass filter with a frequency of 0.1 Hz was used, and the frequency corresponding to the peak amplitude also approached this filter frequency. When we tried to change this high-pass filter frequency using the acceleration sensor, the frequency corresponding to the peak amplitude also moved accordingly. This indicates that high-pass filtering had an important impact on the control system using acceleration sensor, but in practical engineering, the high-pass filtering was necessary. In real spatial payloads, the displacement information of the struts cannot be directly measured and needs to be obtained through a relatively complex inverse solution based on the angular measurement information; moreover, the sampling frequency of the angular measurement information is relatively low, and phase lag occurs, which is not conducive to active control. The strain sensor has a small mass and can directly reflect the vibration information of the single strut. Equipping each strut with a strain sensor makes it easier to implement individual control algorithms for a single strut. Therefore, in this type of APHI system, the strain sensor method fulfills the requirements of an active control sensor and holds promising prospects for aerospace applications.

## 4. Conclusions

This paper presents the design of a micro-vibration hybrid isolation system for space-borne payloads using an electromagnetic damper, piezoelectric actuator, and strain sensor. A corresponding control algorithm based on strain sensing is proposed and the system’s suppression effect is verified through experiments. The key findings of this study are as follows:(1)The designed electromagnetic damper effectively reduces the response at the fundamental frequency of the system. The control algorithm developed using strain sensing achieves effective damping control and minimizes the payload response. The algorithm demonstrates stability according to both theoretical analysis and experimental results.(2)In the hybrid isolation system, the control effect of the acceleration sensor is inferior to those of the displacement and strain sensors, and the control effects of the displacement and strain sensors are comparable. However, the strain sensor method holds more potential to meet the needs of practical aerospace applications than the displacement sensor.

## Figures and Tables

**Figure 1 sensors-24-01649-f001:**
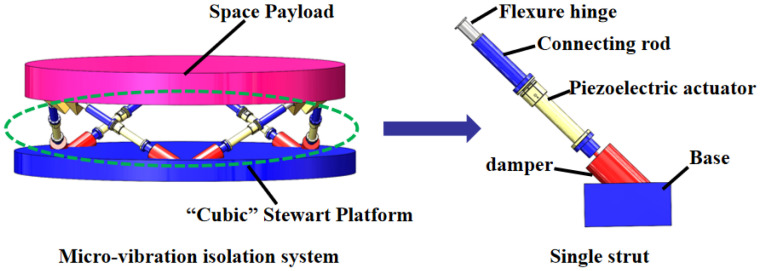
Simplified validation model.

**Figure 2 sensors-24-01649-f002:**
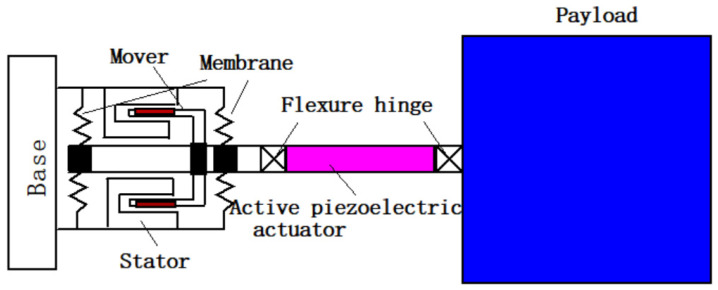
Schematic of the strut structure.

**Figure 3 sensors-24-01649-f003:**
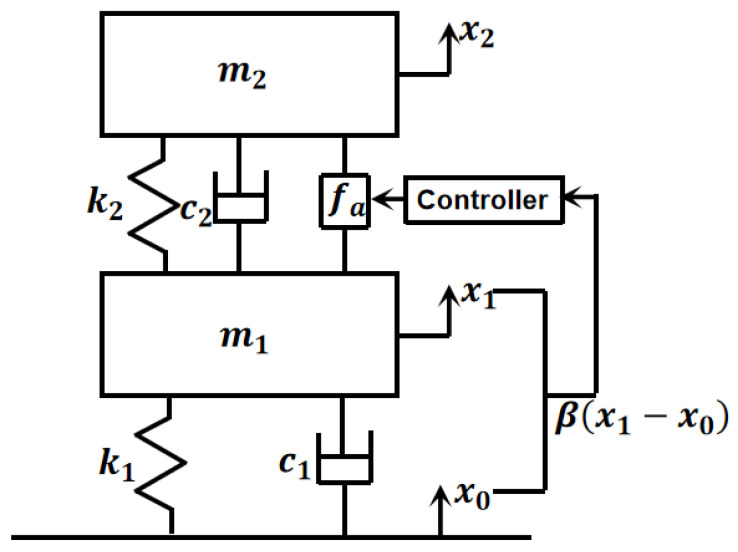
Mathematical model of the single strut.

**Figure 4 sensors-24-01649-f004:**
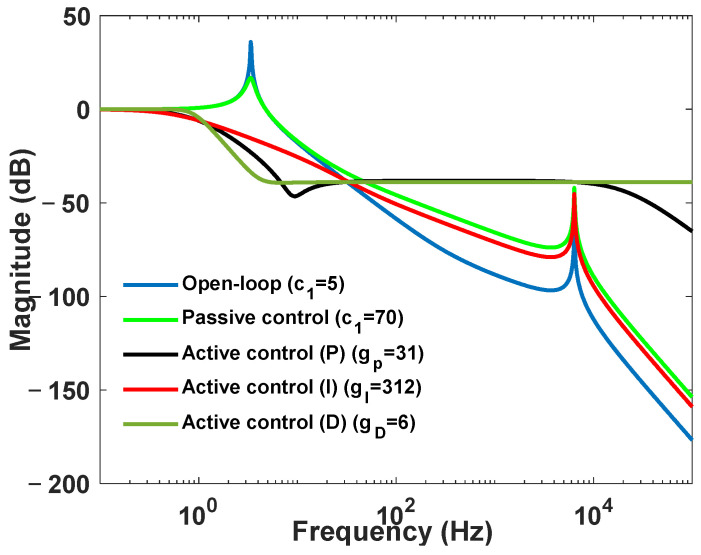
Transmission characteristic diagram.

**Figure 5 sensors-24-01649-f005:**
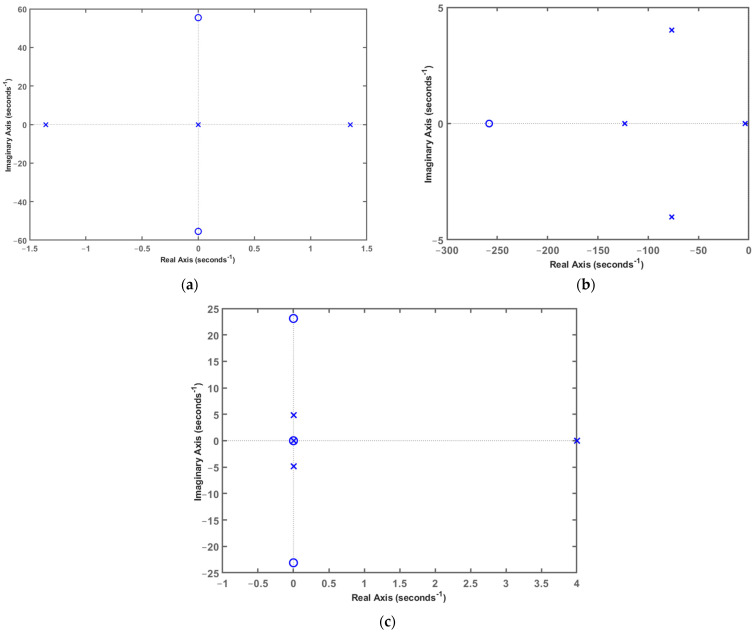
Zero-pole distribution diagram of active control system: (**a**) active control (P); (**b**) active control (I); (**c**) active control (D).

**Figure 6 sensors-24-01649-f006:**
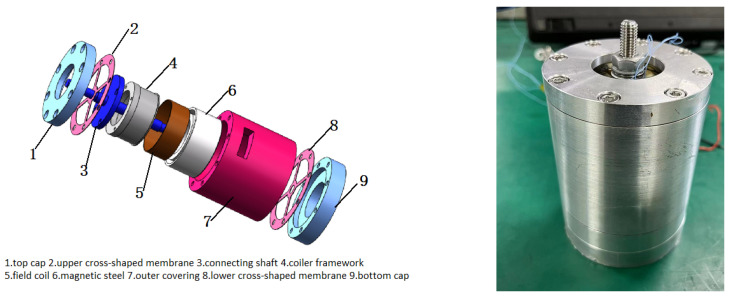
Electromagnetic damper.

**Figure 7 sensors-24-01649-f007:**
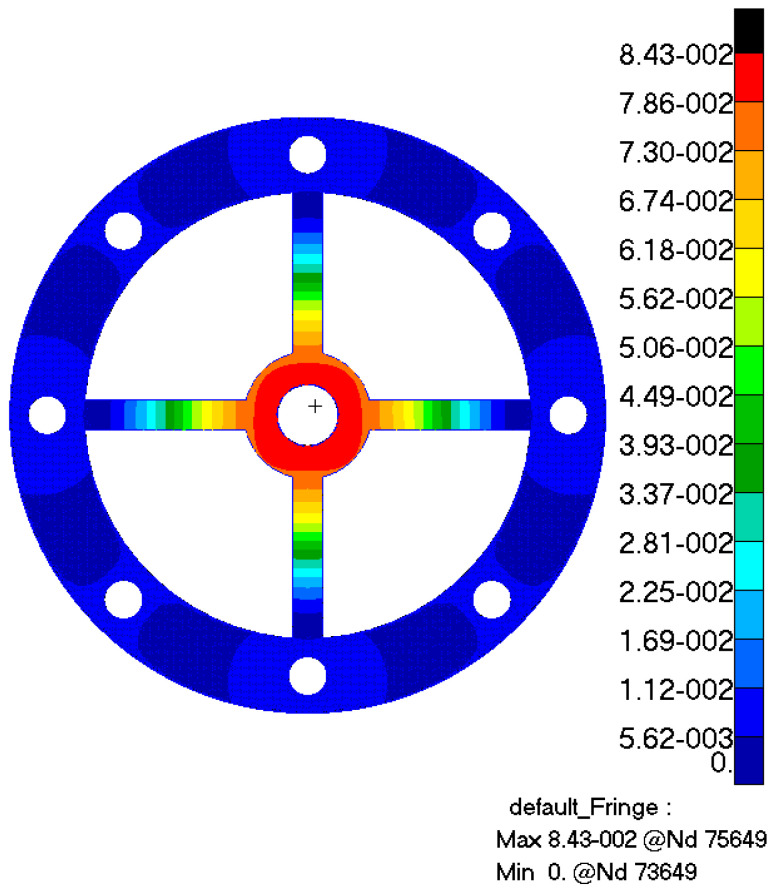
Static analysis.

**Figure 8 sensors-24-01649-f008:**
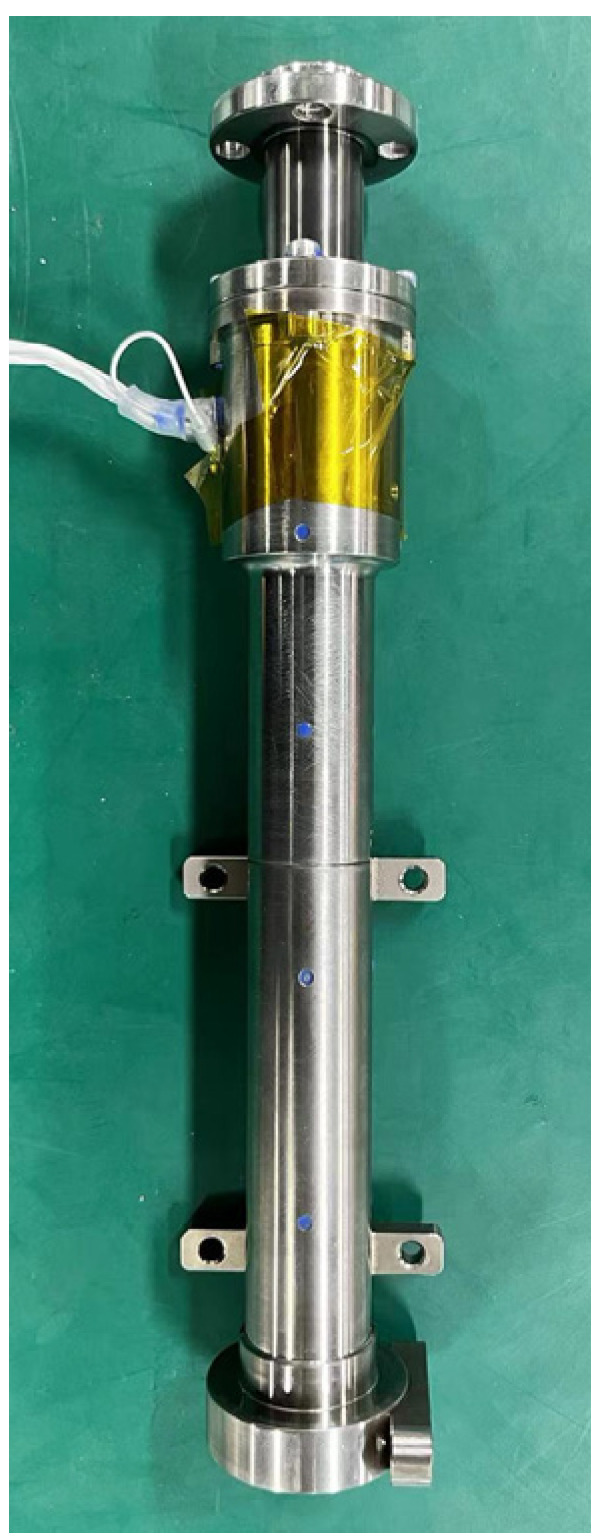
Active piezoelectric actuator.

**Figure 9 sensors-24-01649-f009:**
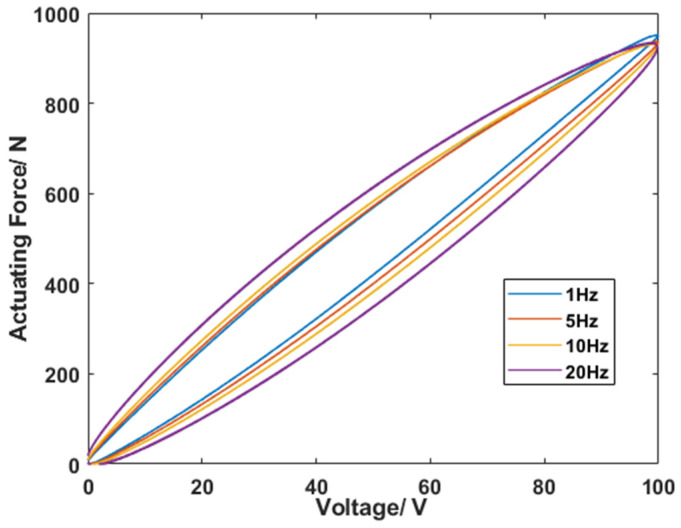
The relationship between actuating force and voltage.

**Figure 10 sensors-24-01649-f010:**
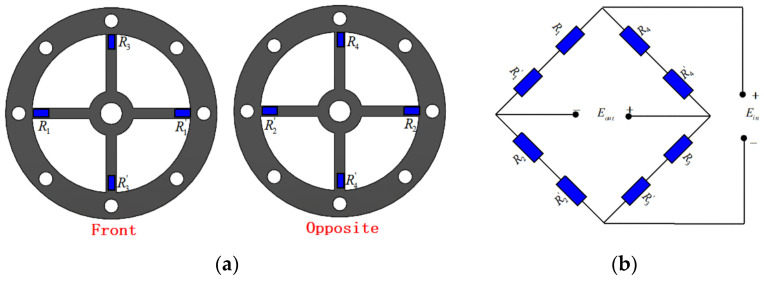
Strain sensing design: (**a**) positions of strain gauges; (**b**) electric bridge.

**Figure 11 sensors-24-01649-f011:**
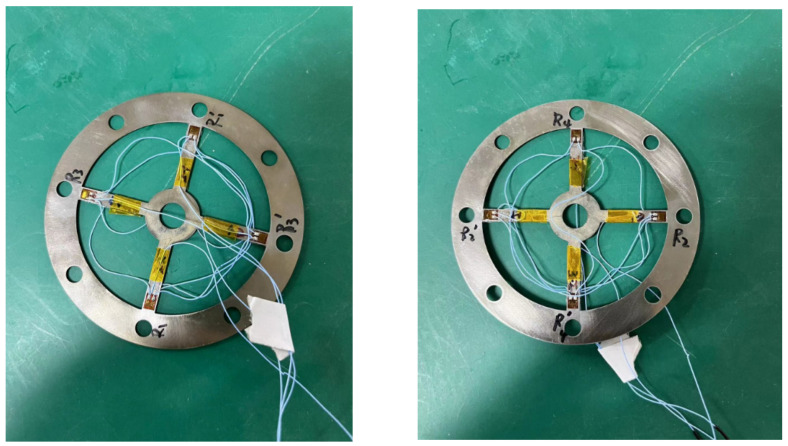
Physical image of strain sensing.

**Figure 12 sensors-24-01649-f012:**
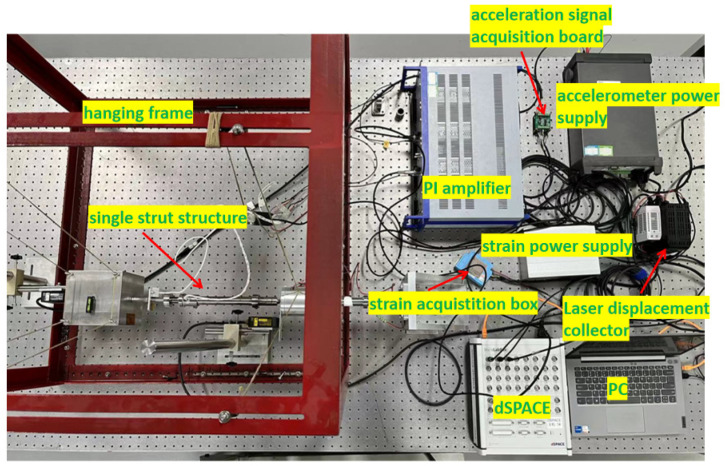
Hybrid vibration isolation experimental system.

**Figure 13 sensors-24-01649-f013:**
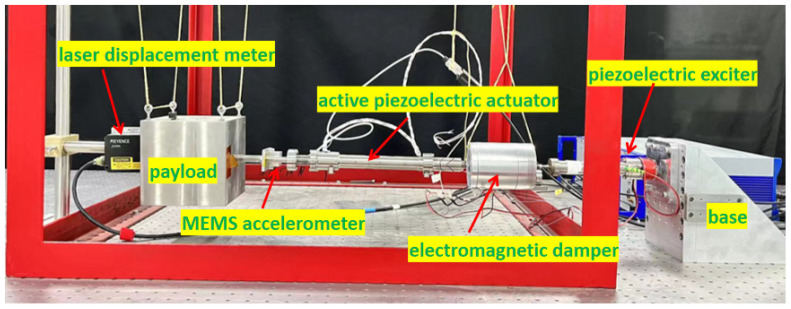
Structural components of the hybrid vibration isolation experimental system.

**Figure 14 sensors-24-01649-f014:**
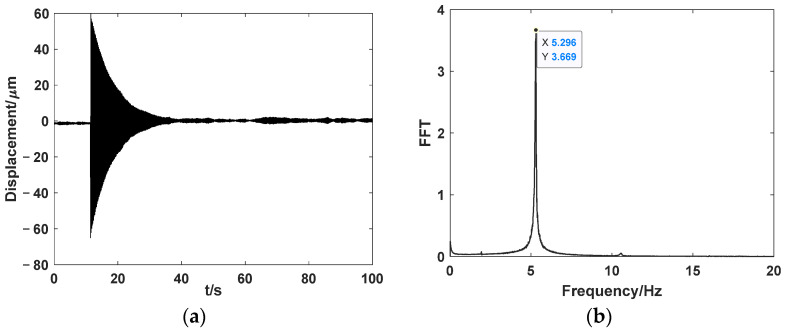
Fundamental frequency test: (**a**) time–domain response; (**b**) FFT.

**Figure 15 sensors-24-01649-f015:**
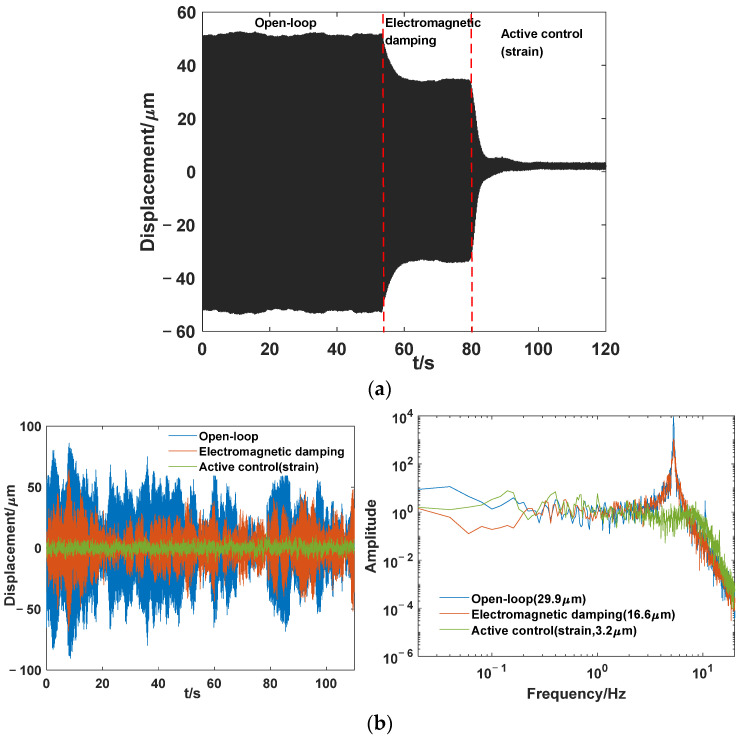
Comparison of the control effects under (**a**) fundamental frequency excitation; (**b**) random excitation.

**Figure 16 sensors-24-01649-f016:**
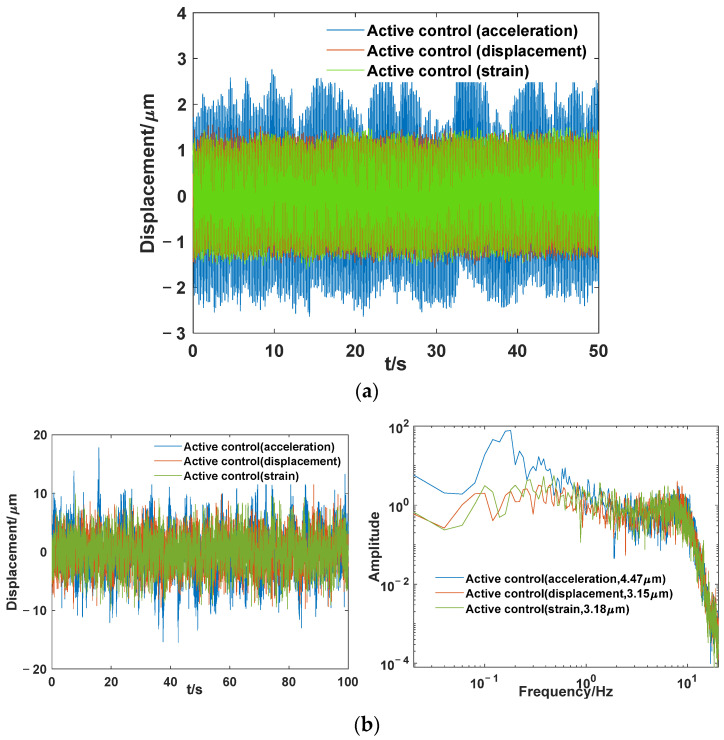
Comparison of three types of sensor control effects under (**a**) fundamental frequency excitation; (**b**) random excitation.

**Table 1 sensors-24-01649-t001:** Parameters of the designed active piezoelectric actuator.

Parameters	Values	Unit
Stroke	±40	μm
Voltage	0~100	V
Tension force	200	N
Thrust force	1000	N
Weight	420	g
Length	241	mm

## Data Availability

Data are contained within the article.

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
