# Peer review of "Design and Experimental Study of a Hybrid Micro-Vibration Isolation System Based on a Strain Sensor for High-Precision Space Payloads"

_sensors, 2024, doi:10.3390/s24051649_

Round 1
Reviewer 1 Report
Comments and Suggestions for Authors
Dear Authors,
The article "Design and Experimental Study of a Hybrid Micro-vibration 2 Isolation System Based on Strain Sensor for High-Precision 3 Space Payloads" is written well. It provides background information on the importance of vibration isolation for space payloads, especially those requiring high precision. However, I will suggest some minor changes before paper acceptance such as:
1) In keywords, Strain is capital rest are small letters; is it a typo or because of some special reason.
2) Reference 4 is just quite old; if possible, remove it as it is just used once combined with [4-7] replace it with some recent year like 2023 and 2024 designs, technologies, and experimental studies in the field, as there is only one citation from last year like https://doi.org/10.3390/mi14071358,https://doi.org/10.1007/s10854-023-10213-3, https://doi.org/10.1016/j.asr.2024.01.020.
3) In "Introduction" section paragraph 2, cite a reference for the frequency band range.
4) Kindly explain how the piezoelectric actuator is responsible for active vibration suppression.
5) Briefly add details about the used membranes, its material selection (benefits of using stainless steel over other materials), from where you got it, and whether there was any difficulty while making the contacts.
6) The piezoelectric actuator was working at what power and maximum frequency. How do you deal with its hanging, which commonly happens when we use these types of actuators.
7) In figures 13 (a), 14 (a), and (b) replace t/s with time/s, as all other parameters have full names.
8) Explain the sharp peaks in Figure 15 (b).
Author Response
Response to Reviewer 1’ Comments
Dear Reviewer,
Thank you very much for editing effort for our manuscript. We have revised our manuscript according to the comments. All the questions are answered point-by-point . If you have any questions about our manuscript, please don’t hesitate to contact us.
Yours sincerely,
Qiwei Guo, Jian Zhou, Liang Li, Minglong Xu, Guoan Tang
Comment 1: In keywords, Strain is capital rest are small letters; is it a typo or because of some special reason.
Response to comment : Many thanks to the reviewer for the careful reading. I am sorry that it is a typo. I have made a revision in the revised manuscript. For more details, please see the highlighted sentences in the revised version.
Comment 2: Reference 4 is just quite old; if possible, remove it as it is just used once combined with [4-7] replace it with some recent year like 2023 and 2024 designs, and experimental studies in the field, as there is only one citation from last year like https://doi.org/10.3390/mi14071358, https: //doi.org /10.1007/s10854-023- 10213- 3, https://doi.org/10.1016/j.asr.2024.01.020.
Response to comment : Many thanks to the reviewer for the revision suggestion. Reference 4 has been replaced using “Shi H. T., Abubakar M. Bai X. T. Vibration Isolation Methods in Spacecraft: A Review of Current Techniques. Advances in Space Research, 2024, https://doi.org/10.1016/j.asr.2024.01.020” in the revised manuscript.
Comment 3: In “Introduction” section paragraph 2, cite a reference for the frequency band range..
Response to comment : Many thanks to the reviewer for the revision suggestion. We have cited a reference [7]” Hanieh A A. Active Isolation and Damping of Vibrations via Stewart Platform. Universite Libre De Bruxelles, 2003” in this part. For more details, please see the highlighted sentences in the revised version.
Comment 4: Kindly explain how the piezoelectric actuator is responsible for active vibration suppression.
Response to comment : Active piezoelectric actuator is designed using piezoelectric stacks. According to the inverse piezoelectric effect of piezoelectric ceramics, piezoelectric actuator can generate corresponding driving forces based on active control signals, and then apply them to the controlled structure to control the movement of the structure, achieving suppression of the controlled object. The source of the active control signal: the real-time sensing signal obtained by the sensor is then calculated through an active control algorithm to obtain the active control signal. For more details, please see the highlighted sentences in the revised version.
Comment 5: Briefly add details about the used membranes, its material selection (benefits of using stainless steel over other materials), from where you got it, and whether there was any difficulty while making the contacts.
Response to comment : Many thanks to the reviewer for the revision suggestion. The main consideration for using stainless steel is that it has a relatively high yield strength and is easy to obtain. There is no difficulty while making the contacts. For more details, please see the highlighted sentences in the revised version. For more details, please see the highlighted sentences in the revised version.
Comment 6: The piezoelectric actuator was working at what power and maximum frequency. How do you deal with its hanging, which commonly happens when we use these types of actuators.
Response to comment : During the experiment, due to the low fundamental frequency of the active-passive system, the active piezoelectric actuator was working at the low-frequency, and the control system performs bandpass filtering from 0.1 to 20Hz. In this case, the power consumption of the piezoelectric actuator is very small. Of course, piezoelectric actuators can operate at higher frequencies.
Sorry, we don’t quite understand this issue “hanging”, and it’s possible we haven’t encountered it. We have developed different piezoelectric actuators, Fast Reflection Mirrors (FSM), and high-precision pointing mechanisms using piezoelectric stacks. We have sufficient experience in this area and can communicate with each other.
Comment 7: In figures 13(a), 14(a), and (b) replace t/s with time/s, as all other parameters have full names.
Response to comment : Many thanks to the reviewer for the revision suggestion. The figures have been modified in the revised manuscript. For more details, please see the highlighted sentences in the revised version.
Comment 8: Explain the sharp peaks in Figure 15(b).
Response to comment : In actual control systems, it is necessary to filter the sensing signal. On the one hand, it removes the direct-current (DC) signal, and on the other hand, it improves the signal-to-noise ratio. In order to remove the DC signal, a high-pass filter with a frequency of 0.1 Hz was used, and the frequency corresponding to the peak value also approached this filter frequency. We tried to change this high-pass filter, and the peak point also moved accordingly. It indicates that high-pass filtering will have an impact on it using acceleration sensor, but in practical engineering, the high-pass filtering is necessary, which is the reason why we need to study strain sensing.

Reviewer 2 Report
Comments and Suggestions for Authors
Comments can be found in the attached file.

Author Response
Dear Reviewer,
Thank you very much for editing effort for our manuscript. We have revised our manuscript according to the comments. All the questions are answered point-by-point . The response to the comments is in the attachment. If you have any questions about our manuscript, please don’t hesitate to contact us.
Yours sincerely,
Qiwei Guo, Jian Zhou, Liang Li, Minglong Xu, Guoan Tang

Reviewer 3 Report
Comments and Suggestions for Authors
This article provided an issue in high-precision space-borne payloads – the impact of micro-vibrations on imaging quality and pointing accuracy. The proposed active–passive hybrid isolation system based on a strain sensor that aims to mitigate these issues. Here are some comments and suggestions for improvement:
(1) Please give a clearly introduction of the significance of the problem and the novelty of the proposed solution.
(2) Should provide more details on the theoretical model used for the hybrid isolation system. Also give a more in-depth explanation of the two-degree-of-freedom system and the integral control method based on strain sensing.
(3) Should describe the manufacturing process of the electromagnetic damper, active piezoelectric actuator, and strain sensor in more detail.
Highlight any challenges faced during the design and manufacturing process and how they were overcome.
(4) Please give the experimental setup conditions and parameters considered during testing.
(5) Should provide sufficient information on the data acquisition process and analysis methods used to evaluate the effectiveness of the proposed system.
(6) Please discuss the criteria used for comparing the control effects of acceleration, displacement, and strain sensors. Explain why strain sensors were chosen and elaborate on the advantages over the other options.
(7) A quantitative analysis of the results, including statistical measures or graphs could be made to support the claims made.
(8) Interpret the results in the context of the initial problem statement and theoretical model. Discuss any discrepancies between the theoretical predictions and experimental findings.
Comments on the Quality of English LanguageDefine any specialized terminology or acronyms to enhance reader understanding.
Author Response

(The authors gave the same response as above.)

Round 2
Reviewer 3 Report
Comments and Suggestions for Authors
Accept as is.
Author Response
Dear Reviewer,
Thank you very much for editing effort for our manuscript again. If you have any questions about our manuscript, please don’t hesitate to contact us.
Yours sincerely,
Qiwei Guo, Jian Zhou, Liang Li, Minglong Xu, Guoan Tang